# Light from van der Waals quantum tunneling devices

Markus Parzefall [1], Áron Szabó[2], Takashi Taniguchi[3], Kenji Watanabe [3], Mathieu Luisier[2] & Lukas Novotny[1]

The understanding of and control over light emission from quantum tunneling has challenged researchers for more than four decades due to the intricate interplay of electrical and optical properties in atomic scale volumes. Here we introduce a device architecture that allows for the disentanglement of electronic and photonic pathways—van der Waals quantum tunneling devices. The electronic properties are defined by a stack of two-dimensional atomic crystals whereas the optical properties are controlled via an external photonic architecture. In van der Waals heterostructures made of gold, hexagonal boron nitride and graphene we find that inelastic tunneling results in the emission of photons and surface plasmon polaritons. By coupling these heterostructures to optical nanocube antennas we achieve resonant enhancement of the photon emission rate in narrow frequency bands by four orders of magnitude. Our results lead the way towards a new generation of nanophotonic devices that are driven by quantum tunneling.

[1] Photonics Laboratory, ETH Zürich, 8093 Zürich, Switzerland. [2] Integrated Systems Laboratory, ETH Zürich, 8092 Zürich, Switzerland. [3] National Institute for Material Science, 1-1 Namiki, Tsukuba 305-0044, Japan. Correspondence and requests for materials should be addressed to M.P. (email: mparzefall@ethz.ch) or to L.N. (email: lnovotny@ethz.ch)

The desire to develop an optical implementation of the classical antenna in its role as a transducer between an electronic source and electromagnetic radiation has revived interest in light emission from inelastic electron tunneling (IET). This process, originally discovered in the 1970s[1], allows for the conversion of electrical energy into optical excitations on length scales considerably smaller than what is achievable with other optoelectronic devices. Recently, IET-driven optical antennas were demonstrated[2,3] and continued development has lead to the achievement of greatly improved device efficiencies[4] and directionality[5], the development of scalable fabrication processes[6,7] as well as the exploration of new applications in sensing[8] and catalysis[9].

The vast majority of IET studies, both on integrated devices[1,10–13] as well as in scanning tunneling microscopy (STM)[14–19], are centered around metal-insulator-metal (MIM) tunnel junctions. In these devices the tunnel junction defines both the electrical circuit and the optical mode confinement, which severely limits the degrees of freedom for device design and optimization. Furthermore, as light emission from MIM devices is the result of a two-step process in which the tunneling electrons first excite a propagating or localized surface plasmon polariton (SPP) mode that subsequently decays via radiation, it is difficult to disentangle the influence of optical and electrical device properties on the light emission process.

A device structure that allows the optical mode to be defined independently of the tunnel junction would provide additional degrees of freedom for device design, adaptability and optimization of performance. Here we introduce and present a first realization of van der Waals quantum tunneling (vdWQT) devices, hybrid van der Waals heterostructures[20,21] comprised of a vertical stack of gold (plasmonic metal), hexagonal boron nitride (h-BN, two-dimensional insulator) and graphene (Gr, two-dimensional semimetal), as illustrated in Fig. 1a. We show that the optical mode confinement in these vdWQT devices can be defined independently of the electrical tunnel junction—setting a new paradigm in interfacing optics and electronics at the nanoscale. As graphene is almost transparent to light; only 2.3% of photons propagating through it are being absorbed[22], we observe—in addition to the two-step process—the direct emission of photons from tunnel junctions that are not coupled to MIM cavities. A comparison to a theoretical model allows us to determine IET to be the source of the observed emission.

We further demonstrate that photon emission from IET can be locally enhanced by coupling to an optical antenna, cf. Fig. 1a. vdWQT device-coupled nanocube antennas emit light from a nanoscopic volume, resulting in the diffraction-limited emission pattern shown in Fig. 1b. Nanocube antennas provide a high local density of optical states (LDOS) and give rise to a narrow emission spectrum, cf. Fig. 1c. On resonance, we achieve an increase of the spectral photon emission rate by more than four orders of magnitude.

To fully understand these results and their implications for future research we first analyze the photon emission from vdWQT devices in the absence of optical antennas before studying the effect of antenna-coupling.

## Results

**Light from uncoupled vdWQT devices.** Figure 2a shows an optical microscope image of a set of devices. They are composed of a vertical stack of a gold electrode, a few-layer h-BN crystal and monolayer graphene (cf. Fig. 1a), assembled on top of a glass (SiO$_2$) substrate. Details on the fabrication process are described in the Methods section. The tunneling devices are formed in the regions where all of the three constituting materials overlap, i.e., at the end of the three left electrodes in Fig. 2a. The electrodes on the right-hand side serve as electrical contacts to the graphene sheet.

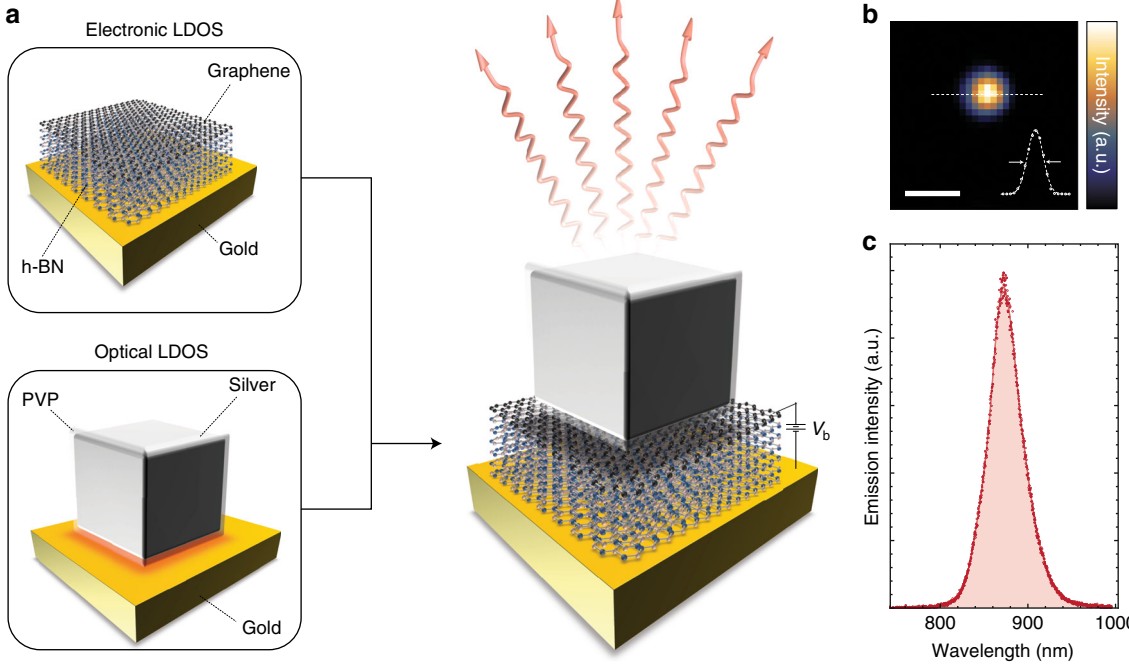

**Fig. 1** Visualization of the vdWQT device concept. **a** Illustration (not to scale) of a gold–few-layer h-BN–graphene vdWQT device, integrated with a (silver, PVP-coated) nanocube antenna. In this device configuration, the electronic LDOS is controlled by the hybrid vdW heterostructure whereas the optical LDOS is governed by the nanocube antenna. Applying a voltage $V_b$ across the insulating few-layer h-BN crystal results in antenna-mediated photon emission (wavy arrows) due to quantum tunneling. **b**, **c** Measured spatial (**b**) and spectral (**c**) photon distribution from a nanocube antenna coupled to a vdWQT device, demonstrating a diffraction-limited spot and a narrow emission spectrum. The inset in **b** shows a line-cut, featuring a line-width (FWHM) of ~460 nm, close to the expected value of $\lambda/(2NA)$ ~ 480 nm. Scale bar: 1 μm

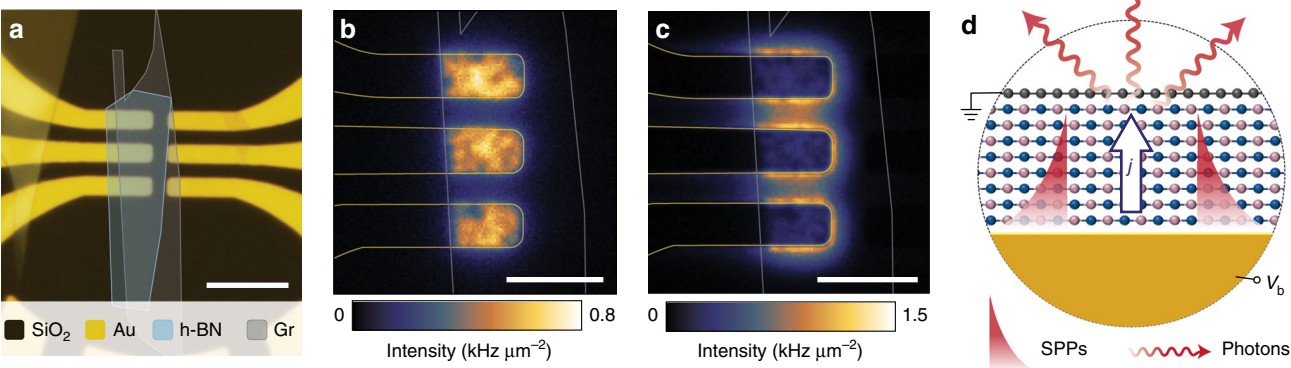

**Fig. 2** Light emission from vdWQT devices. **a** Optical microscope image of a set of devices. The three areas on the left, where gold electrodes (50 nm), h-BN crystal ($N = 7$) and graphene overlap, form the tunneling devices. Electrodes on the right serve as electrical contacts to the graphene sheet. Scale bar: 10 μm. **b** Spatial distribution of photons emitted into the air/upper half-space above the graphene sheet from the devices at an applied voltage of 2.0 V. Light is emitted from the entire area of the devices. Scale bar: 5 μm. **c** Corresponding distribution of photons emitted into to substrate/lower half-space. Light is observed primarily at the edges of the devices. Scale bar: 5 μm. Straight lines in **b**, **c** mark the outlines of gold electrodes (gold) and graphene (gray). Intensity units are as measured and not corrected for system efficiency. **d** Sketch of the vertical cross-section through the device, illustrating the mechanism of light generation. The inelastic component of the tunneling current $j$ couples to the optical modes of the vdWQT device, i.e., the free-space continuum of optical states as well as surface-bound SPP modes (not to scale). Photons emitted into the air half-space give rise to the image in **b**, SPP scattering at the edges of the gold electrodes generate the image in **c**

When a voltage between the graphene and gold electrodes is applied, we observe photon emission into both half-spaces, i.e., the air above the device and the glass substrate. Our analysis regarding the origin of this emission is segmented into two parts. First, we analyze our results in terms of the optical modes that are excited. Second, we will determine which physical mechanism causes the mode excitation.

**Direct and indirect photon emission**. Photons that are emitted into the air half-space give rise to the image shown in Fig. 2b. Evidently, photons are emitted from the entire area of the devices. On the contrary, photons emitted into the glass half-space give rise to the image shown in Fig. 2c, where the emission is localized near the edges of the device. The origin of these images is found by identifying the optical modes of the planar geometry. As we show in Supplementary Note 4, these modes are the radiation continuum of free space and SPP modes bound to the gold surfaces, as illustrated in Fig. 2d (additionally, our geometry supports graphene plasmons which are not considered here, cf. Supplementary Note 4).

On the one hand, tunneling electrons may couple to the radiation continuum, emitting photons without an intermediate excitation. This direct emission process, enabled by the transparency of the top graphene electrode, explains the emission into the top half-space from the device area seen in Fig. 2b. It is important to note that this observation cannot be explained by the random scattering of SPPs by surface roughness as the emission area is restricted to the area of the tunnel junction whereas SPPs are free to propagate across the left edge of the junction area due to the negligible mode mismatch (cf. Supplementary Fig. 3f). On the other hand, photons may also be emitted indirectly as the result of a two-step process. First, a SPP is excited by a tunneling event. Second, since the SPP is bound to the surface it may only emit a photon when it is scattered, that is, at the edges of the gold electrodes. Since photons are preferentially scattered into the medium with the higher refractive index, i.e., glass, we observe photon emission from the edges of the device in Fig. 2c. The relative strength of the coupling to the radiation continuum and the SPP modes is governed by the LDOS associated with each mode. Due to the low

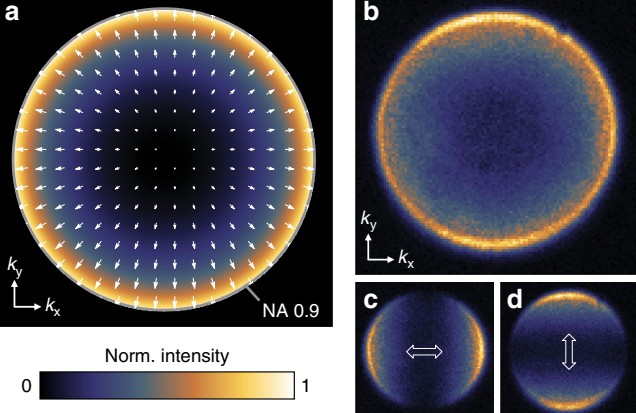

**Fig. 3** Radiation pattern of vdWQT device emission. **a** Calculated emission pattern of an electric point dipole located at a distance of 1 nm above a gold half-space, radiating a free space wavelength of 800 nm. Arrows indicate the position-dependent orientation of the electric field vector. The maximum value of the in-plane wave vector component $k_{||} = (k_x^2 + k_y^2)^{1/2}$ is given by $k_{||,max} = NA \times k_0$ where NA = 0.9 is the numerical aperture of the objective and $k_0 = 2\pi/\lambda$. **b** Measured angular distribution of photons emitted into the upper half-space. **c**, **d** As **b** when measured through a polarizer whose transmitting axis is oriented horizontally/vertically, respectively

mode confinement of the SPP mode the associated LDOS is of the same order of magnitude as the LDOS associated with direct photon emission (cf. Supplementary Note 4), resulting in similar intensities of the two emission pathways, cf. Fig. 2b, c.

To provide further evidence for the direct photon emission process we analyze the angular distribution of photons emitted into the air half-space, shown in Fig. 3. If photons are indeed directly emitted by tunneling electrons their radiation pattern should resemble the radiation pattern of a vertical dipole above a metal surface (cf. Fig. 2d), as calculated in Fig. 3a[23]. The calculation shows that photons are primarily emitted into high angles. Furthermore, as indicated by the white arrows, the

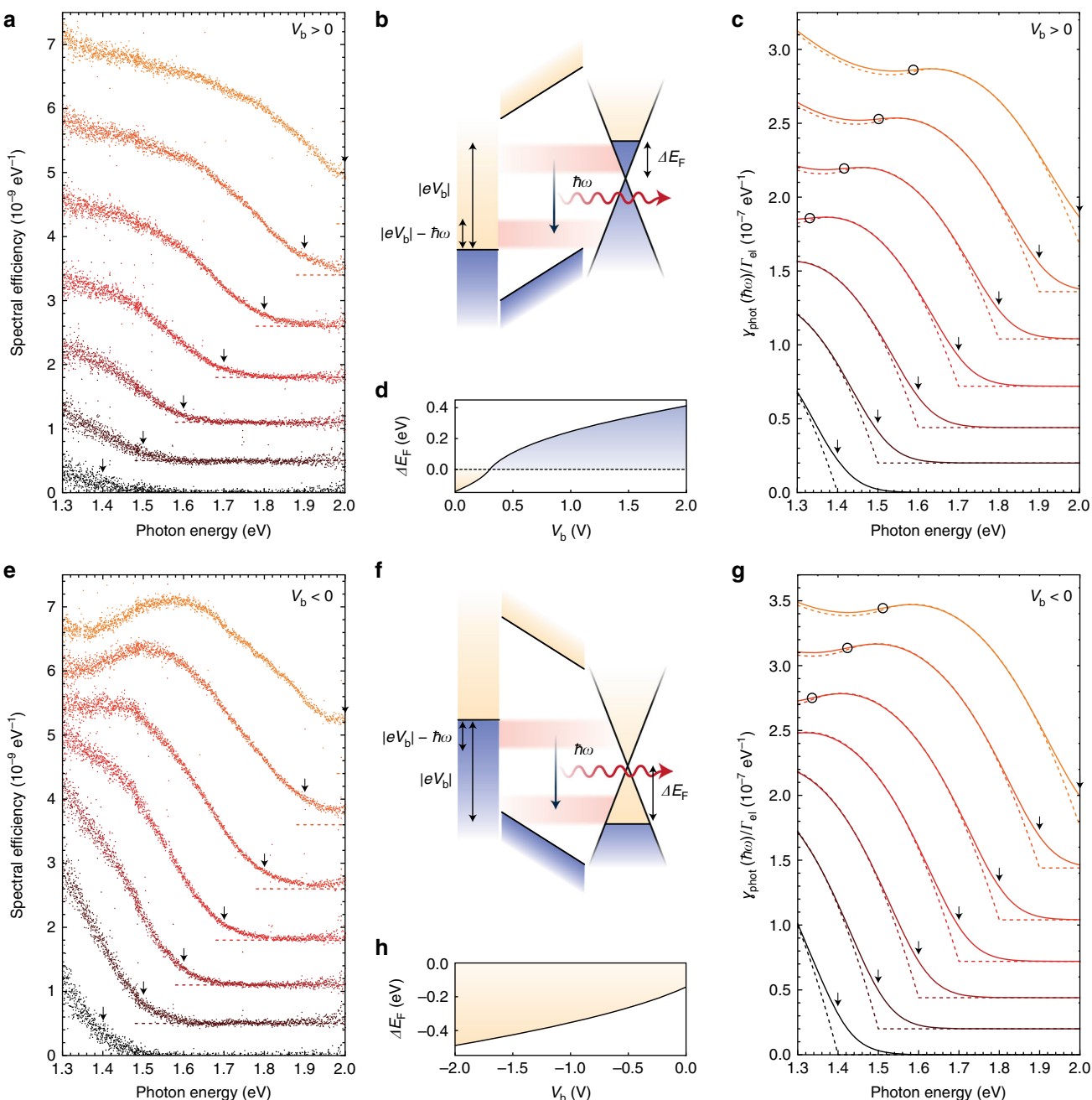

**Fig. 4** Spectral distribution of directly emitted photons from vdWQT devices. **a**, **e** Measured efficiency spectra (spectral distribution of emitted photons per tunneling electron, cf. Supplementary Note 1) for positive/negative voltages $V_b$ from a Au–7L h-BN–Gr device. Subsequent spectra are acquired with increasing voltage from $|V_b| = 1.4$ V (black) to $|V_b| = 2.0$ V (orange) in steps of 0.1 V. Spectra are offset with respect to each other, dashed lines indicate the respective zero line. Arrows indicate the spectral position where $\hbar\omega = eV_b$. **b**, **f** Schematic of inelastic electron tunneling at positive/negative voltages. Optical mode excitations (here: photons) are generated by spontaneous transitions from occupied states in one electrode to unoccupied states in the other electrode. The energy window available for spontaneous excitation of a mode of energy $\hbar\omega$ is given by $|eV_b| - \hbar\omega$. **c**, **g** Theoretical photon emission rate spectra ($\rho_{rad} = 0.04\rho_0$) for positive/negative voltages $V_b$, normalized to the elastic tunneling rate $\Gamma_{el}$. Spectra are calculated for voltages from $|V_b| = 1.4$ V to $|V_b| = 2.0$ V in steps of 0.1 V. Straight and dashed lines show calculations for $T = 300$ K and $T = 0$ K, respectively. Black circles mark the positions on each curve for which the condition $|eV_b| - \hbar\omega = |\Delta E_F|$ is fulfilled. Arrows indicate the spectral position where $\hbar\omega = eV_b$. **d**, **h** Electrostatically induced graphene Fermi level shift $\Delta E_F$ (cf. Methods) as a function of (positive/negative) applied voltage

emission is radially polarized. Experimentally, this distribution is obtained by imaging the back focal plane of the microscope objective that collects the photons. The measured radiation pattern, as well as the decomposition into two orthogonal polarization directions, is shown in Fig. 3b–d. The experimental results stand in excellent agreement with the direct emission mechanism.

**Mode excitation by quantum tunneling**. The remaining question relates to the nature of the emission: What is the physical process that leads to the observed light emission? While it is generally accepted that inelastic electron tunneling—where one[1,15] or multiple[24,25] electrons generate an optical excitation while traversing the barrier—is the dominating mechanism in light emission from MIM tunnel junctions[13,15], it is not a given

that the same is true for the devices presented here due to the two-dimensional nature of the graphene electrode and its unique band structure[26]. The different mechanisms that are discussed in the context of light emission from tunnel junctions are

1. Inelastic electron tunneling.
2. Hot electron photoemission, where the electron first traverses the barrier and subsequently transitions to an energetically lower-lying unoccupied state through the generation of an optical excitation[27], and
3. Thermal emission, where electron tunneling gives rise to a 'hot' electron distribution, which in turn generates black-body-like radiation[28,29].

Irrespective of the mechanism, the spectrum of the light that is emitted is ultimately not only determined by the mechanism itself but also by the frequency-dependence of the LDOS ($\rho_{opt}$) and radiation efficiency ($\eta_{rad}$) of the associated modes in a given sample geometry. In particular, for the case of light emission from MIM tunnel junctions, this interplay of a series of frequency-dependent factors has made it historically difficult to distinguish between the different physical mechanisms that might be at work[13,27]. The direct photon emission from our devices is governed by an optical mode density that is nearly constant in frequency (cf. Supplementary Note 4), allowing us to get an unobstructed view on the underlying mechanisms.

To identify the dominating emission mechanism, we now turn to the spectral distribution of these directly emitted photons, shown for applied voltages in the range from 1.4 to 2.0 V in Fig. 4a, e. First, we find that the spectral efficiency of light emission is very similar for both bias polarities. In MIM devices this finding is in agreement with both inelastic electron tunneling and hot electron photoemission because of the symmetry of the device. The efficiency of hot electron photoemission, that relies on the spontaneous generation of an optical excitation after reaching the counter electrode, depends on the efficacy of competing decay channels. With hot carrier lifetimes in graphene and gold being of the order of 100 fs[30] and 10 fs[31], respectively, one would expect a pronounced bias asymmetry in the emission efficiency, which is not observed. Second, we observe a clear correlation between the spectral cutoff $\hbar\omega_{max}$ and the applied voltage $V_b$ given by $\hbar\omega_{max} \sim |eV_b|$. This indicates that multiple-electron IET processes do not play a dominant role[24,25]. Furthermore, it is not compliant with blackbody-like thermal emission, where the emitted spectrum is determined by the effective temperature of the electron gas rather than the applied voltage[28,29]. Hence, we identify single-electron IET to be the most likely mechanism underlying the observed emission.

**Theoretical device modeling**. To validate this hypothesis we compare our experimental results with theoretical calculations, describing the IET process—as depicted in Fig. 4b, f—as a spontaneous transition between occupied electronic states of one electrode to unoccupied states of the other electrode[13,32–35]. Based on Fermi's golden rule and within the validity range of the dipole approximation, we may express the spectral rate of inelastic electron tunneling as[35]

$$\gamma_{inel}(\hbar\omega) = \frac{\pi e^2}{3\hbar\omega m^2 \varepsilon_0} \rho_{opt}(\hbar\omega) \\ \times \int_{\hbar\omega}^{eV_b} |\mathcal{P}(E, \hbar\omega)|^2 \rho_{Au}(E - \hbar\omega)\rho_{Gr}(E)dE,$$ (1)

where $\rho_{opt}$ is the partial LDOS along the direction of electron tunneling, $\mathcal{P}(E, \hbar\omega)$ is the momentum matrix element (cf. Supplementary Note 2), $\omega$ is the angular frequency, $\hbar\omega$ the energy of the optical excitation and $\rho_{Au/Gr}$ is the electronic density of states of gold/graphene. The equation as stated applies to the positive

voltage range. For negative voltages, the roles of the two electronic state densities are interchanged. The process is schematically depicted in Fig. 4b, f for positive/negative bias. At positive voltages electrons excite optical modes by transitioning from occupied states in the graphene electrode to unoccupied states in the gold electrode. Correspondingly, at negative voltages, optical modes are excited by transitions from occupied states in the gold electrode to unoccupied states in the graphene electrode. At any given mode energy $\hbar\omega$ the energetic range over which inelastic electron tunneling may occur is given by the applied voltage as $|eV_b| - \hbar\omega$. While $\mathcal{P}$ and $\rho_{Au}$ are approximately constant (cf. Supplementary Note 3), $\rho_{Gr}$ depends linearly on $\Delta E_F$, cf. Eq. (4). It is important to realize that the gold layer does not only act as one of the electrodes of the tunnel junction but also as a gate electrode[36,37]. This causes the charge carrier density of the graphene sheet and hence the shift of the Fermi level with respect to the Dirac point ($\Delta E_F$) to depend on the tunnel voltage, as depicted in Fig. 4d, h, cf. Methods. Differences in the band alignment and dipole layers at the gold/h-BN and graphene/h-BN interfaces result in a finite value of $\Delta E_F$ even at $V_b = 0$, cf. Supplementary Note 1.

From Eq. (1) we calculate the spectral photon emission rate as $\gamma_{phot} = \eta_{rad} \times \gamma_{inel} = \rho_{rad} \times \gamma_{inel}/\rho_{opt}$, where $\eta_{rad}$ is the radiation efficiency and $\rho_{rad} = \rho_{opt} \times \eta_{rad}$ is the radiative LDOS. For the planar heterostructure we find that $\rho_{rad} = 0.04\rho_0$, where $\rho_0 = \omega^2 \pi^{-2} c^{-3}$ is the vacuum LDOS and $c$ is the speed of light in vacuum (cf. Supplementary Note 4). The resulting $\gamma_{phot}$ spectra, normalized by the calculated elastic tunneling rate $\Gamma_{el}$ (cf. Supplementary Note 1), are displayed in Fig. 4c, where dashed and straight lines correspond to calculations for $T = 0$ K and $T = 300$ K, respectively. At $T = 0$ K, the quantum cut-off condition $\hbar\omega_{max} = |eV_b|$ is perfectly fulfilled such that the emission rate is zero for energies larger than the applied voltage. At finite temperatures this condition is relaxed by the thermal broadening due to the Fermi-Dirac distribution function, in agreement with experimental results. Most importantly, we are able to reproduce the experimentally observed spectral shape, confirming IET to be the underlying physical mechanism. It is determined by the electronic density of states of graphene, in particular by the voltage-dependent energetic position of the Dirac point (cf. Fig. 4d, h and Methods). Black circles on each curve mark the points for which the condition $|eV_b| - \hbar\omega = |\Delta E_F|$ is fulfilled. This condition determines the spectral location of the observed inflection point. The effect of the density of states minimum is more strongly pronounced for negative voltages. In our modeling we assume a symmetric and energy-independent electronic wave function decay (cf. Supplementary Note 2). Deviations from this assumption, as well as potential effects related to the gate-dependence of the barrier height at the graphene–h-BN interface[38], are possible causes for the observed asymmetry. We emphasize that no fitting parameters are employed in the theoretical model with the exception of the extraction of the residual Fermi level shift at zero applied bias (cf. Supplementary Note 1).

Comparing the absolute values of the experimental and theoretical spectra of Fig. 4a, c, respectively, we find that the experimental efficiencies are lower than the theoretically calculated ones by a factor of ~40. This deviation is caused by the presence of an additional phonon-enhanced tunneling channel in metal–graphene tunnel junctions[39–42], see Supplementary Note 1 for details.

**vdWQT devices coupled to nanocube antennas**. Until now we have analyzed the coupling of tunneling electrons to photons and SPPs of a planar, layered heterostructure—the optical properties

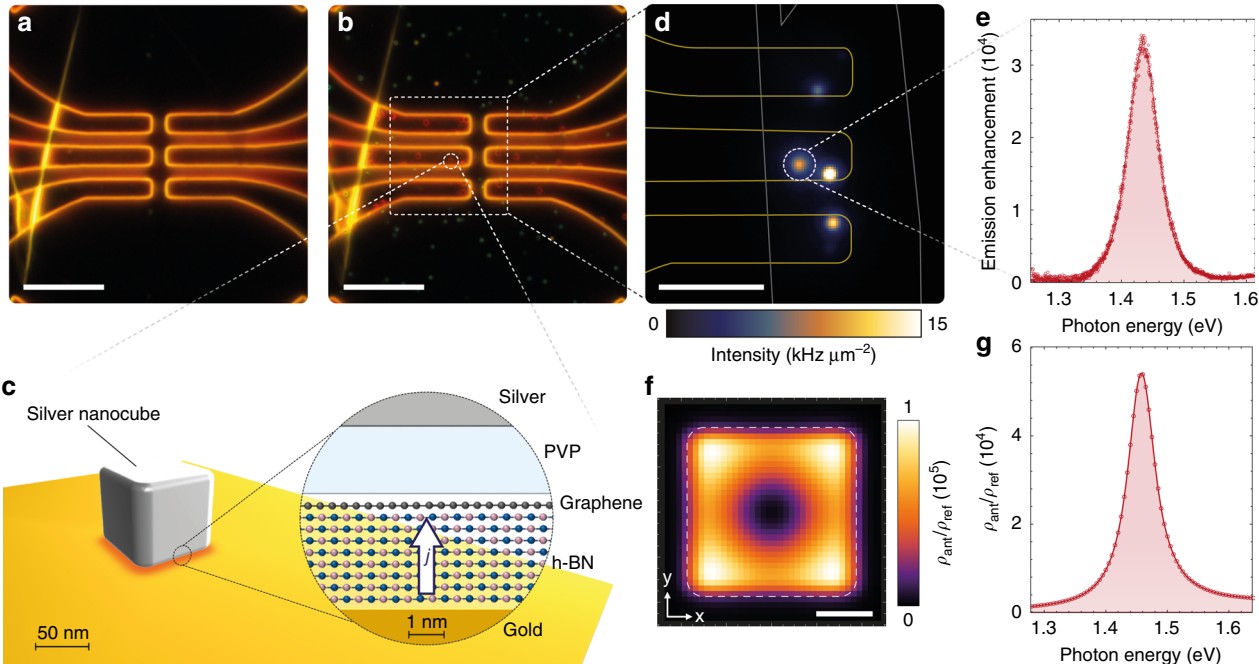

**Fig. 5** Nanocube antennas driven by a vdWQT device. **a** Darkfield microscope image of the same devices as shown in Fig. 2a. Scale bar: 10 µm. **b** Darkfield image after the deposition of silver nanocubes onto the sample. The red ring-like patterns on top of the electrodes indicate the presence of nanocube antennas. Scale bar: 10 µm. **c** Schematic illustration of the nanocube antenna configuration. The inset shows a vertical cross-section of the modified heterostructure configuration which determines the optical modes "seen" by tunneling electrons. **d** Spatial intensity distribution of light emission from nanocube antennas. Outlines indicate the spatial location of the edges of graphene and gold electrodes. Scale bar: 5 µm. **e** Emission spectrum of the antenna marked by a dashed circle in **d** at $V_b = 2$ V, normalized by the corresponding emission spectrum of the planar heterostructure (Fig. 4a). **f** Spatial (on resonance) and **g** spectral dependence of the simulated, radiative fraction of the optical mode density of the nanocube antenna $\rho_{ant}$ for a vertically oriented dipole located in the center of the h-BN crystal, normalized by the radiative LDOS of the planar geometry $\rho_{ref}$, cf. Supplementary Note 4. The white dashed line in **f** marks the outline of the silver nanocube (75 nm with a 7.5 nm edge radius)

of which are characteristic of a single gold film with a nearly energy-independent LDOS. Our findings demonstrate that we can view tunneling electrons as quantum emitters with emission properties that depend on the electronic density of states as well as the optical LDOS, cf. Eq. (1), and can thus be controlled by coupling to optical antennas[43,44] or cavities[45] to increase emission rates and shape emission spectra and emission pattern. To illustrate this concept we couple the vdWQT devices to nanocube antennas[46,47].

Silver nanocubes are deposited on top of the vdWQT devices from solution (cf. Methods) and can be directly observed with a darkfield microscope. Before nanocube deposition, as seen in Fig. 5a, scattering is dominated by the edges of the gold film. After nanocube deposition, as seen in Fig. 5b, we additionally observe randomly distributed spots, which are associated with optical resonances formed in the junctions between nanocubes and gold electrodes, as schematically illustrated in Fig. 5c. It is interesting to note that nanocubes sitting on top of the gold electrodes and the heterostructure produce a red ring-shaped pattern whereas (single) nanocubes sitting on the glass surface appear green. The ring-like pattern is caused by the symmetry of a higher-order MIM resonator mode that dominates the scattering response in the visible spectral range, cf. ref. [48]. The green color of nanocubes on glass originates from the nanocubes dipolar resonance. The distance between the two metallic domains is determined by the number of h-BN layers and the thickness of the PVP coating (~3 nm) that stabilizes the nanocubes in solution.

While the formation of nanocube antennas leaves the electronic properties, i.e., the elastic and phonon-enhanced tunneling channels (c.f. Supplementary Note 1), largely

unaffected—it strongly alters the light emission properties of the device. Without nanocubes we observe emission from the entire device area, cf. Fig. 2b. After antenna-coupling, as shown in Fig. 5d, the spatial intensity distribution is dominated by point-like sources, the size of which is given by the point-spread function of our imaging system. In the following we will focus on and selectively analyze (cf. Methods) the emission spot marked by the dashed circle in Fig. 5d. Further cube antennas are discussed in Supplementary Note 5. Figure 5e shows the photon emission spectrum of this particular nanocube antenna, normalized by the corresponding emission spectrum of the planar heterostructure (Fig. 4a) from an area equivalent to the nanocube footprint (75 × 75 nm², cf. Supplementary Note 5). Previously, before the deposition of nanocubes, the emission spectrum was broad (cf. Fig. 4a) and primarily determined by the electronic density of states of the electrodes, whereas now photons are emitted into a narrow spectral window centered at ~1.43 eV with a full width at half maximum (FWHM) of ~57 meV. Compared to the emission spectrum of the bare vdWQT device, we find that—on resonance —the probability for an electron tunneling event, occurring within the active area of the nanocube antenna, to result in the generation of a photon is enhanced by more than four orders of magnitude. This enhancement is independent of applied voltage and polarity (cf. Supplementary Note 5).

The origin of these alterations is found by analyzing the modified optical properties of the system. According to Eq. (1), the spectral rate of photon emission by inelastic electron tunneling is determined by the optical mode density $\rho_{opt}(\hbar\omega)$. The formation of nanocube antennas modifies $\rho_{opt}(\hbar\omega)$ in two ways. Firstly, SPPs are confined to the nanoscale gap between the silver nanocube and the gold surface. The optical mode density of

these MIM-SPPs is strongly enhanced as a result of the spatial field confinement and the increased propagation constant. Additionally, the termination of the MIM gap by the facets of the cube act as reflectors—forming a resonator—which leads to an additional increase in the mode density by the Purcell effect[49–51]. While $\rho_{opt}$ determines the rate of inelastic electron tunneling $\gamma_{inel}$, the fraction associated with photon emission $\gamma_{phot}$ is determined by the radiative LDOS $\rho_{rad}$.

Classically, the LDOS enhancement $\rho_{opt}/\rho_0$ corresponds to an increase of the power dissipated by a point dipole $P/P_0$ due to its interaction with the environment[23]. This correspondence allows us to determine the LDOS by means of classical electrodynamics (cf. Methods and Supplementary Note 4). The spectral variation of the radiative LDOS of the nanocube antenna $\rho_{ant}$, normalized by the radiative LDOS of the planar geometry $\rho_{ref}$ (cf. Supplementary Note 4), is shown in Fig. 5g. The radiative LDOS is resonantly enhanced at a photon energy of $\hbar\omega_{res} \sim 1.46$ eV and over a spectral range given by the FWHM of ~57 meV. We further analyze the nature of the enhancement by varying the position of the dipole, oscillating at $\omega_{res}$, as shown in Fig. 5f. The lateral variation of $\rho_{rad}$ maps out the fundamental mode of the MIM resonator. The average LDOS enhancement is found to be ~$5 \times 10^4$, in excellent agreement with the experimentally observed photon emission rate enhancement of ~$3 \times 10^4$. These findings validate the dependence of the strength of the inelastic electron tunneling channel on the LDOS experienced by tunneling electrons, as suggested by Eq. (1).

## Discussion

In conclusion, we have introduced and experimentally demonstrated a device platform that allows for the excitation of optical modes at the atomic scale by inelastic electron tunneling—van der Waals quantum tunneling devices. In particular, we have demonstrated the independent control over the electronic and optical properties of the system, both of which determine the spectral rate of inelastic electron tunneling in the form of the respective densities of states, cf. Eq. (1).

The electronic properties and the optical source spectrum are determined by a hybrid van der Waals heterostructure, here: gold–hexagonal boron nitride–graphene. The optical modes of this structure are the photon modes of free space and SPP modes. Based on a careful analysis of the spatial, angular and spectral distribution of the emitted photons, we unambiguously determine the direct emission of light from tunneling electrons to be the source of the observed emission. The emission is well described by a spontaneous emission model, confirming inelastic electron tunneling as its source. The spectral flatness of the optical mode density of the planar heterostructure allows us to observe the influence of the electronic density of states on the emission spectrum.

The optical properties are further modified by an external photonic architecture, here: nanocube antennas. The integration with these optical antennas leads to a spectral shaping and enhancement of the photon emission rate. Nanocube antennas constitute nanoscopic resonators for MIM-SPPs, accompanied by a local enhancement of the LDOS. Additionally, the geometry allows for the efficient coupling of MIM-SPPs to photons, resulting in the observed enhancement of the photon emission rate by more than four orders of magnitude.

This concept is easily translated to more complex systems. To optimize the source efficiency or introduce other device functionalities one may choose materials constituting the van der Waals heterostructure[20,21] from an entire library of two-dimensional atomic crystals[52]. For example, utilizing

semiconducting atomic crystals enables the shaping of the energy-dependence of the electronic density of states and may allow for the suppression of the elastic tunneling channel in favor of the inelastic process. In the devices presented here the source efficiency is reduced compared to traditional MIM tunneling devices due to the dominance of the phonon-assisted tunneling channel (cf. Supplementary Note 1). However, this may be overcome in future device configurations since tunneling between identical atomic crystals may allow for significantly higher efficiencies due to favorable momentum selection rules that have not yet been harnessed in any other material system[53–55].

The individual two-dimensional crystals constituting vdWQT devices very weakly interact with optical modes that are polarized perpendicular to the sample plane as shown here for the case of the long-range SPPs of the planar geometry and the MIM-SPP mode inside the nanocube antenna. The same modes however couple most efficiently to tunneling electrons. This feature of vdWQT devices may be further explored by their integration with optical systems that support corresponding modes such as certain types of photonic cavities[45], hybrid plasmonic waveguide resonators[56,57], or SPP resonators[58–60].

## Methods

**Device fabrication**. Devices are fabricated on commercially available glass coverslips ($22 \times 22 \times 0.13$ mm$^3$). Electrodes are defined by photolithography and electron beam evaporation of 1 nm titanium (Ti, 0.05 nm/s) and 50 nm gold (Au, 0.1 nm/s) at a pressure of $<10^{-7}$ mbar. Lift-off is performed sequentially in acetone, isopropyl alcohol and deionized water. The RMS roughness of the gold electrodes is 0.4–0.5 nm.

Graphene and few-layer h-BN are mechanically exfoliated from bulk crystals onto oxidized silicon wafers (100 nm SiO$_2$). h-BN crystals were grown as described in ref. [61]. Suitable crystals are identified optically. Their thickness, i.e., the number of layers, is determined by a combination of optical contrast and atomic force microscopy. h-BN crystals with thicknesses of 6–7 layers (corresponding to ~2–2.3 nm) are chosen as a compromise between the voltage that may be applied without reaching the electrostatic breakdown field of ~1 V/nm and the tunneling current which decreases exponentially with crystal thickness[62,63]. To assemble the tunneling devices, graphene and few-layer h-BN are picked up successively with a polydimethylsiloxane (PDMS)/polycarbonate (PC) stack. The vdW stack is subsequently transferred on top of the gold electrodes. For pickup and transfer we follow the procedure developed by Zomer et al.[64]. The PC film employed as a carrier is dissolved in chloroform after successful transfer. Finally, the samples are annealed at 200 °C in a tube furnace while flowing 380 sccm argon and 20 sccm hydrogen to reduce residues on top of the heterostructure.

Nanocube antennas are assembled by drop-casting 50 µL of silver nanocubes (size: 78 ± 5 nm) in ethanol (concentration approximately 0.1 mg/mL, purchased from Nanocomposix) on top of the gold–h-BN–graphene assembly. After a short incubation time of several seconds the sample is spun dry at ~3000 rpm.

Following this approach we have assembled ten gold–h-BN–graphene heterostructures with varying numbers of individual devices and nanocube antennas, the properties of which are in agreement with the representative set of experimental data and their evaluation shown in this work.

**Optical and electrical device characterization**. The devices are characterized on a customized, inverted Nikon TE300 microscope under ambient conditions. Electrical measurements are carried out using a Keithley 2602B source meter unit. Light emitted from the devices into the air half-space is collected by an air objective (Nikon, 100×, NA 0.9) whereas light emitted into the glass half-space is collected by an oil-immersion objective (Nikon, 100×, NA 1.4). For real space and Fourier space (backfocal plane) imaging, the respective plane is imaged onto a electron-multiplying charge-coupled device (EMCCD, Andor iXon Ultra). The emission from individual nanocube antennas is selectively analyzed by spatial filtering in an intermediate image plane using a pinhole with a diameter of 100 µm, corresponding to a diameter of 1 µm in the sample plane. Spectra are acquired using a Princeton Instruments Acton SpectraPro 300i spectrometer. Spatial intensity distribution maps as shown in Figs. 2b, c and 5d are not corrected for system transmission, detection efficiency and the point spread function of the imaging system that determines the size of the emission spots in Fig. 5d. Spectral efficiency and photon flux spectra as shown in Figs. 4a, e and 5e and Supplementary Figs. 4a, 5a are corrected for the transmission function of the optical system and detection efficiency. They are further normalized to the area from which the photons are emitted, i.e., the entire device area in Fig. 4a and the footprint of the nanocube antenna in Fig. 5e as well as Supplementary Figs. 4a, 5a.

**Device electrostatics**. The dependence of the charge carrier density of graphene ($n_{Gr}$) on the applied voltage can be described as

$$e(V_b - V_0) = \frac{e^2 n_{Gr}}{c_g} + \Delta E_F, \qquad (2)$$

where $c_g = \varepsilon_0 \varepsilon_{hBN}/d_{hBN}$ is the geometric capacitance per unit area, $\varepsilon_0$ is the vacuum permittivity and $\varepsilon_{hBN}$ and $d_{hBN} = N \times d_{il}$ are the DC permittivity and thickness of the h-BN crystal, respectively, with $N$ being the number of h-BN layers and $d_{il} = 0.33$ nm being the h-BN interlayer distance[65]. The second term, $\Delta E_F$, accounts for the quantum capacitance of the two-dimensional graphene sheet[66,67]. The charge carrier density is related to the electronic density of states ($\rho_{Gr}$) through

$$n_{Gr} = \int_0^{\Delta E_F} \rho_{Gr}(E) dE. \qquad (3)$$

Near the Dirac point energy $E_D$, $\rho_{Gr}$ depends linearly on energy as[26]

$$\rho_{Gr}(E) = \frac{2|E - E_D|}{\pi \hbar^2 v_F^2}, \qquad (4)$$

where $v_F$ is the Fermi velocity of graphene.

Combining Eqs. 2–4 results in a quadratic equation in $\Delta E_F$ that allows us to explicitly calculate $\Delta E_F$ as a function of voltage. For calculations, we assume $v_F = 1.1 \times 10^6$ m/s[68] and $\varepsilon_{hBN} = 3.5$[69].

**Local density of optical states**. The inelastic electron tunneling rate, described in Eq. (1), depends on the LDOS $\rho_{opt}$. For uncoupled vdWQT devices we calculate the LDOS semi-analytically (Supplementary Note 4). The LDOS of antenna-coupled vdWQT devices is calculated numerically (Supplementary Note 4). In both approaches we make use of the fact that the normalized LDOS $\rho_{opt}/\rho_0$ is equivalent to the normalized power $P/P_0$ dissipated by a classical point dipole with moment $\mathbf{p}$[23].

## Data availability

The data that support the findings of this study are available from the corresponding author upon reasonable request.

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

## Acknowledgements

The authors thank A. Jain, N. Flöry, S. Heeg, M. Frimmer and D. Svintsov for valuable discussions. This work was supported by the Swiss National Science Foundation (grant no. 200021_165841) and the ETH Zürich (ETH-32 15-1). The authors further acknowledge the use of facilities at the FIRST Center for Micro- and Nanotechnology at ETH Zürich. K.W. and T.T. acknowledge support from the Elemental Strategy Initiative conducted by the MEXT, Japan and the CREST (JPMJCR15F3), JST.

## Author contributions

M.P. and L.N. conceived the device concept. M.P. fabricated and characterized the devices. M.P. performed all analytical calculations and numerical simulations. Á.Sz. and M.L. performed ab initio device simulations. K.W. and T.T. synthesized the h-BN crystals. M.P. and L.N. discussed results and co-wrote the paper with contributions from Á.Sz. and M.L.

## Additional information

**Competing interests:** The authors declare no competing interests.

