## [Peer Review File · Nature Communications]

Reviewers' Comments:

Reviewer #1:

Remarks to the Author:

In the manuscript: "Light from Van der Waals quantum tunneling devices" the authors experimentally realize tunneling based light emission from graphene-hBN-Au. They further show plasmon enhancement from Ag nanocubes deposited on the surface of the structure. I believe this is an interesting study, with comprehensive theoretical analysis, and therefore I recommend publication. I have a few minor comments for the authors to consider:

1) The LDOS is much less than in vacuum. I do not fully understand this since my understanding is that a metal film will typically enhance the LDOS. Physically, LDOS to me does not correspond to enhanced collection, but rather just enhanced emission rate. I am assuming that the calculation that the authors do is for enhanced emission into the far-field (correspond to the collection efficiency) and not for enhanced emission overall. In any case, this part was confusing. Probably a clear distinction between radiative and total LDOS should be made in the text.

2) Cube number 5 seemed to show brighter emission in the CCD image than 4, but was spectrally weaker. I am assuming that this is because of some spectral efficiencies of the camera. Perhaps some discussion about this discrepancy should be given.

3) More discussion about the absolute efficiencies would be appreciated. Considering Figure 4 and Figure 5, it seems that even with the Ag cube, the efficiency is about 10^{-5} . This is seemingly less than previous works using STM emission from wires by some of the same authors.

4) How does the Ag cube influence the tunneling process? It could be that it enhances the tunneling at that point or perhaps screens it out. My sense is that it would provide some capacitive enhancement.

Reviewer #2:

Remarks to the Author:

The paper reports a study of light emission by inelastic tunneling through a BN film between a graphene and a gold electrode. It is shown that by placing silver nanocubes on top of the graphene, the light emission can be locally enhanced at the frequency corresponding to the nanocube resonance. This is an important result in a field where many groups are currently working. The paper is very well written. I recommend publication after revision.

The experiment are very well described and there is an excellent discussion. In particular, the discussion of the mechanism of light emission by inelastic tunneling is very convincing and the analysis of the signal accounting for the electronic density of states of graphene is particularly interesting. To my knowledge, this is an original part of the paper. The effect of the nanocubes is very clear experimentally and deserves to be published. However, the mechanism requires a more detailed discussion. I will recommend publication after clarification of some issues raised below.

1) The definition of four orders of magnitude enhancement is unclear. The paper does not provide a clear definition of the enhancement. It seems to be performed by taking the ratio of two spectra. However, this is probably dependent on the numerical aperture of the objective used to collect the data. If this is the case, this enhancement is not an intrinsic quantity and is of no interest. The agreement with the LDOS enhancement may be accidental.

What is the field of view ? is it larger than the nanocube ? by how much ? More information is needed to understand what is this enhancement. Ideally, the field of view should match the absorption cross section of the nanocube.

2) Related to the previous comment is the following question: Fig. 2c shows that light emission in the substrate is mediated by plasmons scattered by the edges. Hence, the very same plasmons could be scattered resonantly by the nanocubes at a particular frequency in air. In that case, light emission enhancement would result from a different mechanism: increased radiation coupling of the already emitted plasmons. This possible mechanism should be discussed. It is not obvious that it can be ruled out. Light emission in air is attributed to direct emission as opposed to scattering from inspection of figure 2b but when inserting resonant scatterers with a size on the order of 70 nm is a very different situation.

3) Light emission in the air due to plasmons is ruled out from fig.2b where it is seen that light is only emitted from the regions where BN and graphene are deposited. This argument implies that plasmons would be transmitted through the interface between plasmons on gold/air and plasmons on gold/BN/graphene/air. This implicit assumption should be discussed. A total reflection may take place at this interface for most angles of incidence.

4) Another interesting information would be to compare the total amount of light emitted from the sample. As the area where the enhancement takes place is very small, I expect a negligible increase of the total emitted light. Is this correct ?

5) In the supplementary material, it is indicated that some nanocubes do not emit light. I am not yet convinced by the explanation of the observation of dark nanocubes based on locally suppressed tunneling because tunneling does not depend on the cube. I acknowledge that the proposed explanation of locally perturbed tunneling at the graphene -BN contact is consistent with the proposed mechanism of direct light emission.

Yet, one could also argue that if light emission by the nanocubes is due to resonant scattering of plasmons, dark nanocubes could result from non illuminated nanocubes.

Images of plasmons usually display speckle fluctuations. Here, the broad emission spectrum and the large distribution of sources may reduce the speckle contrast. I suggest to the authors to take the same picture as in figure 2b using a filter at 1.45 eV with a bandwidth on the order of 60 meV equivalent to the width of the nanocube. I expect the contrast of the speckle pattern to increase as a consequence of the smaller bandwidth. Hence, darker regions might appear. If a nanocube is in one of these dark spots, it will not be illuminated and will therefore not scatter light. That alternative explanation could also explain the existence of dark spots.

An interesting indication could be spectral images of the type of Fig. 2 b. If light emission is due to plasmons scattering, the pattern should change when changing frequency observation. If it is only due to tunneling inhomogeneity and direct light emission, the pattern should be the same for all frequencies.

Reviewer #1 (Remarks to the Author):

In the manuscript: "Light from Van der Waals quantum tunneling devices" the authors experimentally realize tunneling based light emission from graphene-hBN-Au. They further show plasmon enhancement from Ag nanocubes deposited on the surface of the structure. I believe this is an interesting study, with comprehensive theoretical analysis, and therefore I recommend publication. I have a few minor comments for the authors to consider:

We thank the reviewer for her/his positive assessment of our work. We will address the reviewer's comments in the following.

1) The LDOS is much less than in vacuum. I do not fully understand this since my understanding is that a metal film will typically enhance the LDOS. Physically, LDOS to me does not correspond to enhanced collection, but rather just enhanced emission rate. I am assuming that the calculation that the authors do is for enhanced emission into the far-field (correspond to the collection efficiency) and not for enhanced emission overall. In any case, this part was confusing. Probably a clear distinction between radiative and total LDOS should be made in the text.

We fully agree with the reviewer that the apparent suppression of the LDOS is surprising at first sight, as one would expect the (radiative) LDOS to increase in the proximity of a metallic surface. The suppression however is a result of the increased refractive index at the position of the emitter and can be understood when analyzing a z-oriented dipole inside a homogeneous medium of refractive index n . Mathematically, considering eq. (S16), the angular spectrum of the dipole cuts off at $s = n$. For $s > n$ the integrand becomes imaginary. When evaluating the integral from 0 to n we find that the LDOS increases linearly with n as $\rho_n = n \rho_0$. However, in our experimental configuration, we can only detect the emission in the range of $s = [0, NA]$. Again, turning to equation (S16) we find that when $\epsilon = n^2$ increases, the value of the integral over $s = [0, NA]$ decreases. As the refractive index of h-BN is ~ 2 , this decrease is rather substantial and causes the suppression of the LDOS in this angular range.

We further thank the reviewer for pointing out the confusion between radiative and total LDOS in this context. For the planar heterostructure we define the (detectable, for simplicity) radiative LDOS to be equivalent to the angular spectrum corresponding to $s = [0, NA] = [0, 0.9]$.

Changes to the manuscript:

- In the "Theoretical device modeling" section, we now define γ_{inel} as the total spectral rate of inelastic electron tunneling, which is given by the partial LDOS along the direction of electron tunneling, ρ_{opt} . In the next paragraph we then introduce γ_{phot} as the spectral photon emission rate, which is the experimentally relevant quantity that we want to compare with our measurements, given by $\gamma_{\text{inel}} \eta_{\text{rad}}$.
- We have added a footnote in the discussion of the radiative LDOS of the planar heterostructure in the supplementary information discussing the origin of the low value of the radiative LDOS

2) Cube number 5 seemed to show brighter emission in the CCD image than 4, but was spectrally weaker. I am assuming that this is because of some spectral efficiencies of the camera. Perhaps some discussion about this discrepancy should be given.

This apparent discrepancy is caused by two factors. First, as correctly suggested by the reviewer, the spectral efficiency of the camera and also the transmission of optical elements between the emitter and the camera decrease with decreasing photon energy. Second, chromatic aberrations in the near-IR range of the imaging objective make it impossible to bring all of the emitting cubes in focus at once. Hence the image shown in Fig. 5d is a compromise in this regard.

Changes to the manuscript:

- We have added a footnote in Section S5 of the Supplementary Information discussing the apparent discrepancy in brightness.

3) More discussion about the absolute efficiencies would be appreciated. Considering Figure 4 and Figure 5, it seems that even with the Ag cube, the efficiency is about 10^{-5} . This is seemingly less than previous works using STM emission from wires by some of the same authors.

Indeed, the overall electron-to-photon conversion efficiencies are of the order of 10^{-5} . This is the same order of magnitude as we demonstrated previously in slot-antenna-coupled tunnel junctions (Nature Nanotech. **10**, 1058 (2015)). The reason is that while the optical properties of the nanocube antennas are most certainly superior to slot antennas, it is the additional phonon-assisted tunneling channel, which we analyze in detail in the Supplementary Section S1, that reduces the efficiency by a factor of 40. The momentum-mismatch between electrons in the graphene and gold electrodes strongly favors phonon-assisted tunneling, which is associated with a large momentum transfer. We have previously not addressed this topic explicitly in the manuscript as this is not the main focus of our work. However, overcoming this limitation is our current research focus. As mentioned in the Discussion section, tunneling devices featuring two identical atomic crystals as electrodes are theoretically predicted to strongly favor inelastic electron tunneling over elastic tunneling while still being integrable with e.g. nanocube antennas.

Changes to the manuscript:

- We have modified one paragraph in the Discussion section (added sentence is underlined): *This concept is easily translated to more complex systems. To optimize the source efficiency or introduce other device functionalities one may choose materials constituting the Van der Waals heterostructure from an entire library of two-dimensional atomic crystals. For example, utilizing semiconducting atomic crystals enables the shaping of the energy-dependence of the electronic density of states and may allow for the suppression of the elastic tunneling channel in favor of the inelastic process. In the devices presented here the source efficiency is reduced compared to traditional MIM tunneling devices due to the dominance of the phonon-assisted tunneling channel (cf. Supplementary Sect.~S1). However, this may be overcome in future device configurations since tunneling between identical atomic crystals may allow for significantly higher efficiencies due to favorable momentum selection rules that have not yet been harnessed in any other material system.*

4) How does the Ag cube influence the tunneling process? It could be that it enhances the tunneling at that point or perhaps screens it out. My sense is that it would provide some capacitive enhancement.

We thank the reviewer for raising this important question. Our experimental observation is that the cube does not significantly alter the local tunneling behavior except for its enhancement of the inelastic tunneling channel of course. While we cannot infer this from the IV-characteristics directly as the cube only affects a vanishingly small fraction of the entire junction area, we can analyze its effect in terms of the voltage-dependent emission characteristics. As pointed out by the reviewer, the cube could potentially locally modify the capacitance and hence alter the gating characteristics / the dependence of the graphene Fermi level on the applied voltage. If the cube would have a strong effect on the local gating characteristics, its enhancement of the photon emission rate would depend on the applied voltage. We have now included our analysis of this voltage-dependence for four cubes in Supplementary Section S5.2 including a new Fig. S5. While there are some deviations, considering that the cube probes an area that is several orders of magnitude smaller than the area of the entire junction without cubes, the agreement shows that there is no strong influence of the cube. This also makes sense as the cube is separated from the graphene only by three nanometers of insulating PVP, which allows for a charge equilibration between the silver and the graphene. One cause of the minor deviations we see could be a local change in the graphene Fermi level offset V_0 , which can result from the cubes' attachment to graphene.

Changes to the manuscript:

- One additional Supplementary Figure S5 with the accompanying discussion in Supplementary Section S5.2.

Reviewer #2 (Remarks to the Author):

The paper reports a study of light emission by inelastic tunneling through a BN film between a graphene and a gold electrode. It is shown that by placing silver nanocubes on top of the graphene, the light emission can be locally enhanced at the frequency corresponding to the nanocube resonance. This is an important result in a field where many groups are currently working. The paper is very well written. I recommend publication after revision.

The experiment are very well described and there is an excellent discussion. In particular, the discussion of the mechanism of light emission by inelastic tunneling is very convincing and the analysis of the signal accounting for the electronic density of states of graphene is particularly interesting. To my knowledge, this is an original part of the paper. The effect of the nanocubes is very clear experimentally and deserves to be published. However, the mechanism requires a more detailed discussion. I will recommend publication after clarification of some issues raised below.

We thank the reviewer for her/his positive assessment of our work and the tentative recommendation to publish. In the following we respond to all the points raised.

1) The definition of four orders of magnitude enhancement is unclear. The paper does not provide a clear definition of the enhancement. It seems to be performed by taking the ratio of two spectra. However, this is probably dependent on the numerical aperture of the objective used to collect the data. If this is the case, this enhancement is not an intrinsic quantity and is of no interest. The agreement with the LDOS enhancement may be accidental.

The enhancement factor is indeed defined as the ratio of two spectra. Experimentally, we observe an enhancement in the photon emission spectrum and compare the spectra of photons that are emitted from the footprint of the nanocube antenna in the presence and absence of the antenna. What we show then is that this enhancement very well agrees with the enhancement of the radiative LDOS. In this regard we have to admit that, in the previous version of the manuscript, we did not clearly discuss the difference between the total and the radiative LDOS. We now distinguish between two rates. On the one hand, we define the spectral rate of inelastic electron tunneling γ_{inel} , which is determined by the partial LDOS ρ_{opt} along the direction of electron tunneling. On the other hand, we define the spectral photon emission rate γ_{phot} , which is determined by the radiative LDOS ρ_{rad} . We base our definition on the radiative LDOS of a planar heterostructure and evaluate the radiation emitted into the angular range that we can detect with our NA=0.9 objective. This corresponds to the integral over the angular spectrum from $s=0$ to $s=0.9$ (cf. Supplementary Sect. S4.1 and Fig. S3a/b). It captures more than 50% of the LDOS corresponding to radiation into the air half-space. Our intention with this definition is to describe our experiment as well as possible and not to artificially generate large enhancement factors. For the nanocube antenna-coupled heterostructure we again determine the average radiative LDOS across the footprint of the antenna, i.e. the fraction that results in the emission of photons. We then take the ratio between the two to describe the radiative LDOS enhancement. The agreement between the local photon emission rate enhancement and the LDOS enhancement is by no means accidental but demonstrates clearly the dependence of photon emission from inelastic tunneling on the radiative LDOS.

Changes to the manuscript:

- In the “Theoretical device modeling” section, we now define γ_{inel} as the total spectral rate of inelastic electron tunneling, which is given by the partial LDOS along the direction of electron tunneling, ρ_{opt} . In the next paragraph we then introduce γ_{phot} as the spectral photon emission rate, the experimentally relevant quantity that we want to compare with our measurements, given by $\gamma_{\text{inel}} \eta_{\text{rad}}$.
- We have rephrased the definition of the experimental enhancement factor in the section “VdWQT devices coupled to nanocube antennas”. The corresponding part now reads as: *Figure 5e shows the photon emission spectrum of this particular nanocube antenna, normalized by the corresponding emission spectrum of the planar heterostructure (Fig. 4a) from an area equivalent to the nanocube footprint (75 x 75 nm²).*

What is the field of view ? is it larger than the nanocube ? by how much ? More information is needed to understand what is this enhancement. Ideally, the field of view should match the absorption cross section of the nanocube.

In the absence of nanocubes we measure the light from the entire device area. After nanocube deposition we selectively analyze the emission from individual nanocubes by spatial filtering, as described in the Methods section. Experimentally this is achieved by generating an intermediate real space image of the emission. In this intermediate image, which is magnified by a factor of 100, we place a 100 μm diameter pinhole that restricts our field of view to a circular area of 1 μm diameter. This allows us to selectively analyze the emission from a single cube. Unfortunately, we cannot filter out the photon emission from the area of the tunnel junction which is still present around the nanocubes. However this contribution can be neglected as the emission from the antenna-coupled junction is four orders of magnitude higher than the background emission while the ratio of the nanocube area (75 x 75 nm²) to the field of view area ((0.5 μm)² π) is only about two orders of magnitude.

Relevant changes to the manuscript:

- We have included more specific information as to how we carry out the spatial filtering in the Methods section – Optical and electrical device characterization, which now reads as: *The emission from individual nanocube antennas is selectively analyzed by spatial filtering in an intermediate image plane using a pinhole with a diameter of 100 μm , corresponding to a diameter of 1 μm in the sample plane.*
- The main text now directly refers to the Methods section when starting the discussion of the nanocube antenna-coupled emission analysis. The corresponding sentence now reads as: *In the following we will focus on and selectively analyze (cf. Methods) the emission spot marked by the dashed circle in Fig. 5d.*

2) Related to the previous comment is the following question: Fig. 2c shows that light emission in the substrate is mediated by plasmons scattered by the edges. Hence, the very same plasmons could be scattered resonantly by the nanocubes at a particular frequency in air. In that case, light emission enhancement would result from a different mechanism: increased radiation coupling of the already emitted plasmons. This possible mechanism should be discussed. It is not obvious that it can be ruled out.

Light emission in air is attributed to direct emission as opposed to scattering from inspection of figure 2b but when inserting resonant scatterers with a size on the order of 70 nm is a very different situation.

We thank the reviewer for pointing out this possible alternative mechanism. To answer this question we carry out the following estimations using our analysis of the cube antenna analyzed in the main text. The emission enhancement on resonance is 3×10^4 , meaning the photon emission rate within the footprint of the antenna ($75 \times 75 \text{ nm}^2$) is enhanced by this factor. On the other hand, the area of the center electrode ($\sim 8 \mu\text{m}^2$) is $\sim 1.4 \times 10^3$ times larger. Hence, on resonance, the total photon emission rate of the planar electrode is a factor of 20 smaller than the photon emission rate of the cube antenna. However, the LDOS associated with the emission of SPPs is roughly a factor of three larger than the radiative LDOS (at 1.43 eV, cf. Supplementary Fig. S3e). Hence, the SPP generation rate across the entire electrode is a factor of seven lower than the photon emission rate of nanocube antenna.

To further estimate the fraction of SPPs that are scattered by nanocube antennas we conducted additional numerical simulations to determine their SPP “scattering width” (which has units of length), which we find to be approximately half the nanocube edge length (power scattered by cube antenna divided by power/unit length injected into the SPP mode) on resonance. As SPPs are emitted into all directions at any given position across the gold-h-BN-graphene junction area we may determine the average scattering efficiency as follows. At any given position, the probability for an emitted SPP to be scattered by the nanocube is given by the ratio of the circumference of a circle whose radius is given by the distance between the position and the cube to the scattering length. Following this approach we find the scattering efficiency to be less than 1%. Hence we conclude that the contribution of SPP scattering is two to three orders of magnitude lower than what is measured experimentally and can thus be safely neglected.

Relevant changes to the manuscript:

- We have added Supplementary Sect. S5.3, discussing this potential alternative mechanism in detail.

3) Light emission in the air due to plasmons is ruled out from fig.2b where it is seen that light is only emitted from the regions where BN and graphene are deposited. This argument implies that plasmons would be transmitted through the interface between plasmons on gold/air and plasmons on gold/BN/graphene/air. This implicit assumption should be discussed. A total reflection may take place at this interface for most angles of incidence.

This is a good point, but we believe that this scenario can be safely ruled out. First, the restriction of the emission to the area of the tunnel junction is not the only indicator for the direct photon emission mechanism. An even stronger argument is the direct correspondence of the angular distribution to a z-oriented dipole (Fig. 3).

Second, the left termination of the junction area only determines the edge of the graphene electrode (Fig. 2a). The mode mismatch between Au-h-BN-Graphene-Air and Au-h-BN-Air is negligible as shown in Supplementary Fig. S3f. Furthermore, the mode index mismatch between Au-hBN-Graphene-Air and Au-Air is less than 1% across the entire relevant spectral range. As the reflection coefficient is related to the mode index mismatch at the interface we find it to be negligible.

Relevant changes to the manuscript:

- We now explicitly mention the negligible mode mismatch as mentioned above and refer to the corresponding Supplementary Figure in the main text, subsection “Direct and indirect photon emission”. The corresponding sentence now reads as: *It is important to note that this observation cannot be explained by the random scattering of SPPs by surface roughness as the emission area is restricted to the area of the tunnel junction whereas SPPs are free to propagate across the left edge of the junction area due to the negligible mode mismatch (cf. Supplementary Fig. S3f).*

4) Another interesting information would be to compare the total amount of light emitted from the sample. As the area where the enhancement takes place is very small, I expect a negligible increase of the total emitted light. Is this correct ?

In fact, the two contributions are roughly equal. As can be seen from Fig. 5d, the antenna-coupled emission spots (where photons emerging from the nanoscale footprint of the antenna are distributed across diffraction limited spots) are well distinct from the background emission (Fig. 2b). Hence, despite the negligible area covered by the nanocubes, the added photon output is of the same order of magnitude as the photon emission from the entire uncoupled junction area.

5) In the supplementary material, it is indicated that some nanocubes do not emit light. I am not yet convinced by the explanation of the observation of dark nanocubes based on locally suppressed tunneling because tunneling does not depend on the cube. I acknowledge that the proposed explanation of locally perturbed tunneling at the graphene -BN contact is consistent with the proposed mechanism of direct light emission. Yet, one could also argue that if light emission by the nanocubes is due to resonant scattering of plasmons, dark nanocubes could result from non illuminated nanocubes. Images of plasmons usually display speckle fluctuations. Here, the broad emission spectrum and the large distribution of sources may reduce the speckle contrast. I suggest to the authors to take the same picture as in figure 2b using a filter at 1.45 eV with a bandwidth on the order of 60 meV equivalent to the width of the nanocube. I expect the contrast of the speckle pattern to increase as a consequence of the smaller bandwidth. Hence, darker regions might appear. If a nanocube is in one of these dark spots, it will not be illuminated and will therefore not scatter light. That alternative explanation could also explain the existence of dark spots. An interesting indication could be spectral images of the type of Fig. 2 b. If light emission is due to plasmons scattering, the pattern should change when changing frequency observation. If it is only due to tunneling inhomogeneity and direct light emission, the pattern should be the same for all frequencies.

We thank the reviewer for their suggestion. Following our previous discussion (point #2) we can exclude the possibility of dark cubes originating from “non-illuminated” cubes as the contribution from SPP scattering to the observed signal is negligible. Also, the image shown in Fig. 2b shows the direct emission of photons from the tunnel junctions and hence does not entail any information about potential SPP speckle fluctuations. The correspondence of the angular distribution of light emitted into the air half-space with a z-oriented dipole as well as the restriction of the emitting area to the actual tunnel junction area provide strong support in favor of the direct emission mechanism, which cannot be reconciled with SPP scattering. We also note that, as the mechanism of inelastic electron tunneling is a spontaneous emission process, the individual tunnel events and hence SPP generation events are incoherent. They hence do not display features that rely on a fixed phase-coherence amongst them. Any residues that are trapped between the layers of the VdW heterostructure (due to the fabrication process) are likely to

suppress tunneling locally, and hence give rise to variations in the brightness of the planar heterostructure (Fig. 2b), as well as amongst nanocube antennas.

Reviewers' Comments:

Reviewer #1:

Remarks to the Author:

The authors have suitably addressed my concerns. I recommend publication.

Reuven Gordon

Reviewer #2:

Remarks to the Author:

I have read the detailed answer of the authors.

I recommend publication in the present form of the manuscript.